

# Viability and management of the Asian elephant (*Elephas maximus*) population in the Endau Rompin landscape, Peninsular Malaysia

Salman Saaban[1], Mohd Nawayai Yasak[1], Melvin Gumal[2], Aris Oziar[2], Francis Cheong[3], Zaleha Shaari[4], Martin Tyson[5] and Simon Hedges[5]

[1] Department of Wildlife and National Parks (DWNP), Ministry of Water, Land and Natural Resources, Kuala Lumpur, Malaysia
[2] Malaysia Program, Wildlife Conservation Society (WCS), Kuching, Sarawak, Malaysia
[3] Johor National Parks Corporation (JNPC), Kota Iskandar, Johor, Malaysia
[4] Department of Town and Country Planning (DTCP), Kuala Lumpur, Malaysia
[5] Global Conservation Programs, Wildlife Conservation Society (WCS), Bronx, NY, United States of America

Corresponding author
Simon Hedges,
simonhedges@asianarks.org

## ABSTRACT

The need for conservation scientists to produce research of greater relevance to practitioners is now increasingly recognized. This study provides an example of scientists working alongside practitioners and policy makers to address a question of immediate relevance to elephant conservation in Malaysia and using the results to inform wildlife management policy and practice including the National Elephant Conservation Action Plan for Peninsular Malaysia. Since ensuring effective conservation of elephants in the Endau Rompin Landscape (ERL) in Peninsular Malaysia is difficult without data on population parameters we (1) conducted a survey to assess the size of the elephant population, (2) used that information to assess the viability of the population under different management scenarios including translocation of elephants out of the ERL (a technique long used in Malaysia to mitigate human–elephant conflict (HEC)), and (3) assessed a number of options for managing the elephant population and HEC in the future. Our dung-count based survey in the ERL produced an estimate of 135 (95% CI [80–225]) elephants in the 2,500 km$^2$ area. The population is thus of national significance, containing possibly the second largest elephant population in Peninsular Malaysia, and with effective management elephant numbers could probably double. We used the data from our survey plus other sources to conduct a population viability analysis to assess relative extinction risk under different management scenarios. Our results demonstrate that the population cannot sustain even very low levels of removal for translocation or anything other than occasional poaching. We describe, therefore, an alternative approach, informed by this analysis, which focuses on in situ management and non-translocation-based methods for preventing or mitigating HEC. The recommended approach includes an increase in law enforcement to protect the elephants and their habitat, maintenance of habitat connectivity between the ERL and other elephant habitat, and a new focus on adaptive management.

## INTRODUCTION

Asian elephants (*Elephas maximus*) are declining in the wild as a result of habitat loss, fragmentation, and degradation; illegal killing (e.g., for ivory and other products or in retaliation for crop depredations); and in some countries removal of elephants from the wild (*Blake & Hedges, 2004*; *Choudhury et al., 2008*; *Leimgruber et al., 2003*). Peninsular Malaysia still has relatively extensive tracts of tropical forest that are habitat for elephants, tigers (*Panthera tigris*), and other endangered species but agricultural expansion (including forest monoculture plantations) is probably the most significant threat to these large mammals in Malaysia (*Clements et al., 2010*). Such expansion is not new: large tracts of lowland dipterocarp forests have been converted to agricultural plantations as a result of both government and private land development schemes since the early twentieth century (*Aiken & Leigh, 1985*; *Khan, 1991*). The land area under oil palm plantations in particular has increased dramatically at the expense of elephant habitat. For example, from 1990 through 2005, 55–59% of oil palm expansion in Malaysia originated from the clearance of natural forests (*Koh & Wilcove, 2008*). By the time of this study, approximately 27% of Peninsular Malaysia was covered by rubber and oil palm plantations and small-holdings, with approximately the same total area covered by these crops in 2018 (Malaysian Palm Oil Board data for September 2011 and 2018 and Annual Rubber Statistics for 2010 and 2018 from the Malaysian Department of Statistics). The expansion of industrial-scale agriculture and forest plantations resulted in a large increase in human–elephant conflict (HEC) not least because oil palm and rubber are frequently eaten or damaged by elephants, resulting in very large financial losses for plantation owners (*Zafir, Wahab & Magintan, 2016*). Small-scale village agriculture is also vulnerable to crop depredations by elephants. In addition to such HEC, the fragmentation and loss of elephant habitat increases the ease of access for poachers and disrupts elephant movements, ultimately leading to the creation of small isolated populations (*Clements et al., 2010*).

As the area under rubber, oil palm, and other plantation crops expanded, particularly as a result of major land development initiatives beginning in the 1910s and 1960s, the most frequent approach to dealing with HEC was to kill the elephants. For example, between 1967 and 1977, 120 crop-raiding elephants were killed (*Khan, 1991*). Starting in 1974, however, the Department of Wildlife and National Parks (DWNP) began implementing an alternative strategy known as translocation, which involves the capture and removal of elephants from conflict areas and their subsequent release in a small number of protected areas, especially Taman Negara. Between 1974 and 2005, DWNP translocated 527 elephants (*DWNP, 2006*). Despite the best of intentions, the dense forest and difficult terrain in the release sites generally prevented post-release monitoring and thus an evaluation of the translocation program. However, two elephants (one male, one female) were fitted

with satellite telemetry collars and the subsequent monitoring revealed that translocated elephants do not necessarily remain within release sites. For example, the adult female released in Taman Negara left that national park and ranged erratically over an area of almost 7,000 km$^2$ (*Stüwe et al., 1998*). Moreover, in addition to the uncertain outcomes of the translocation program, it is expensive, involves dangers for both people and elephants, and perhaps most significantly, the impact of capturing and removing elephants on the source populations themselves is poorly known.

There is, therefore, a need to consider alternatives to translocation and more generally to better incorporate elephant conservation into national development strategies, especially land use planning, as part of Malaysia's strategy of balancing development and conservation. This need is perhaps most clear in the southern part of the Malaysian peninsula, including in the Endau Rompin Landscape (ERL), where significant changes in land use are currently in progress or at the planning stage with the potential for significant increases in HEC as well as the loss of connectivity between areas of wildlife habitat. The ambitious Central Forest Spine (CFS) plan of the Malaysian Government aims to maintain such connectivity but to be successful needs to be informed by up to date information on the distribution of elephants and other wildlife distribution (*DTCP, 2009*).

The ERL comprises Endau Rompin State Park (in Pahang State), Endau Rompin Johor National Park (Johor State), and large areas of Permanent Reserve Forest in Johor and Pahang States that are connected to the two parks (Fig. 1). The ERL covers an area of about 3,600 km$^2$, contains what is believed to be one of the three most important elephant populations in Peninsular Malaysia, and contains a CITES[1] Monitoring the Illegal Killing of Elephants (MIKE) program site. The ERL is located within a matrix of other land cover types, especially oil palm and rubber plantations to the north, west, and south. The presence of these plantations adjacent to elephant habitat has led to high levels of HEC and significant numbers of elephants have been translocated out of the ERL as a result (*DWNP, 2006*).

The objectives of our study were, therefore, to provide up to date information on the elephant population in the ERL (because such data were lacking) and to use those data to help improve the conservation and management of the species. Specifically, we conducted a survey to assess the size of the elephant population (which was unknown), used that information to assess the viability of the population (which was believed to be closed geographically) under a number of management scenarios especially those involving translocation, and then assessed a number of options for managing the elephant population and HEC in the future. More generally, the need for conservation scientists to produce research of greater relevance to practitioners is now increasingly recognized (*Arlettaz et al., 2010*; *Cook, Hockings & Carter, 2009*; *Laurance et al., 2012*; *Meijaard & Sheil, 2007*; *Meijaard, Sheil & Cardillo, 2014*). We aimed therefore to provide a concrete example of scientists working alongside practitioners and policy makers to address a question of immediate relevance to wildlife conservation in Malaysia (i.e., the size and viability of a key elephant population and its vulnerability to offtake including translocation and poaching) and then to use the results to inform wildlife management policy and practice in Malaysia.

[1] Convention on International Trade in Endangered Species of Wild Fauna and Flora.
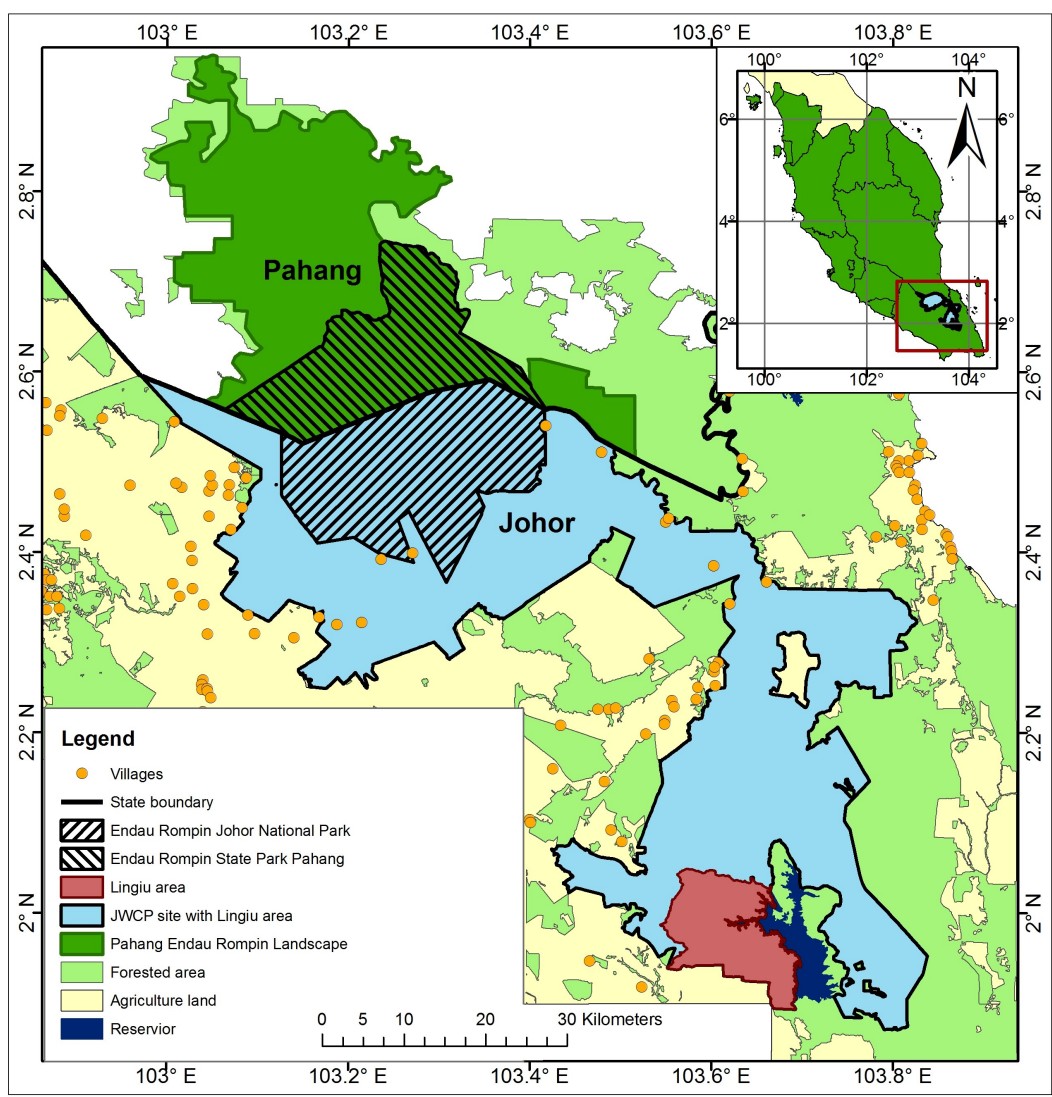

**Figure 1** **Map of Peninsular Malaysia showing the location of the Endau Rompin Landscape (ERL).** The ERL comprises the areas identified as ''Pahang Endau Rompin Landscape'' plus the ''JWCP site with Lingui area'' and the ''Lingui area''; the total area of the ERL is c. 3,600 km$^2$ and it is entirely within Pahang and Johor States.

## MATERIALS & METHODS

### Study area

We used our knowledge of elephant ecology in conjunction with topographic maps, vegetation cover data, and land use data for the ERL, information from our earlier reconnaissance work in the ERL, data from others working in the area, and DWNP data including HEC data to delimit plausible boundaries for the area occupied by the elephant population in the ERL. Thus, for example, large areas of peat swamp were excluded as was the Lingiu Development Zone (Fig. 1). The resulting study area covered c. 2,500 km$^2$

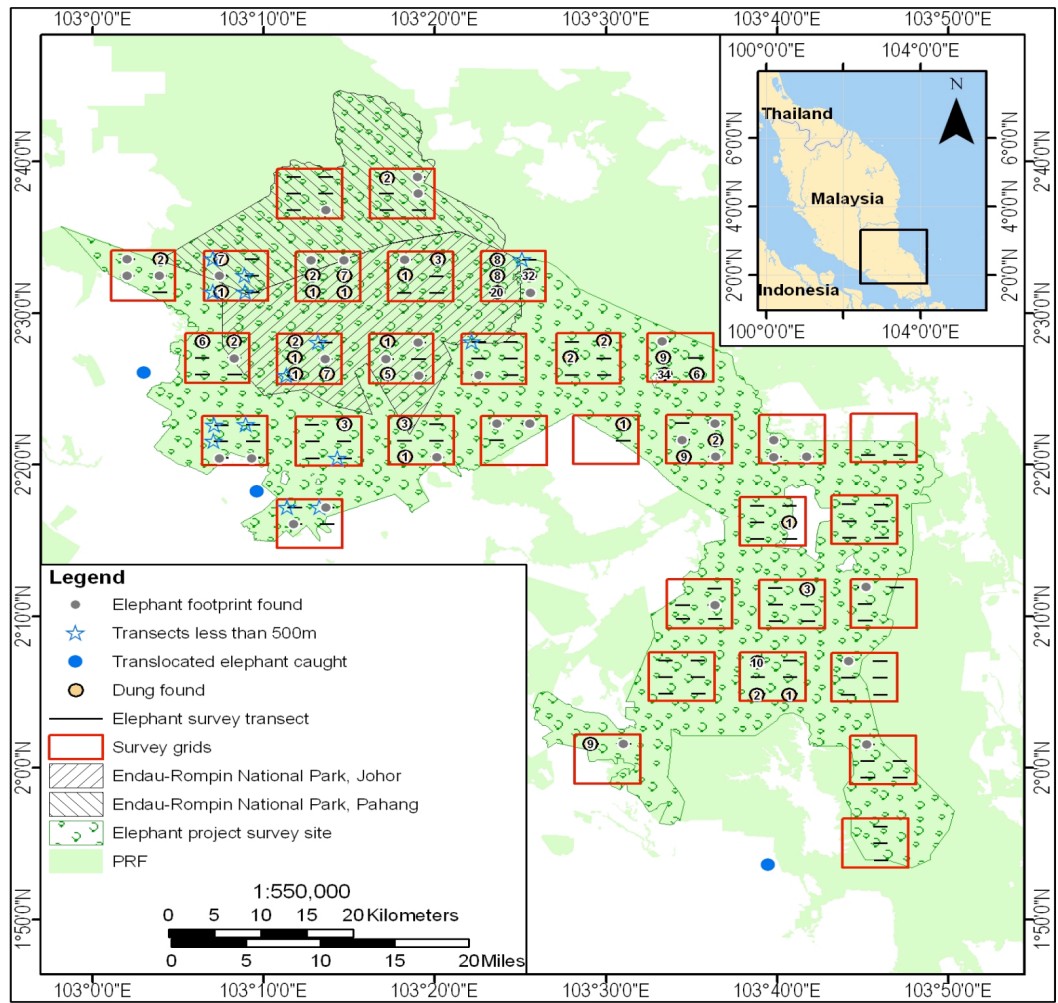

**Figure 2** **Location of the line transects used for the 2008 survey of the 2,500 km² elephant study area in the Endau Rompin Landscape.** Transects are shown as horizontal black lines; the numbers within the orange circles indicate the number of dung piles found per transect.

and included Endau Rompin State Park (Pahang State), Endau Rompin Johor National Park (Johor State), the CITES MIKE site (Mersing District, Johor State), and a large area of Permanent Reserve Forest in Johor State not included in either the park or the MIKE site (Fig. 2). The forest in the protected areas comprises mixed dipterocarp forest of the Keruing–Red Meranti (*Dipterocarpus shorea*) and Kapur (*Dryobalanpus*) types (*Wong, Saw & Kochummen, 1987*). During the 1970s and 1980s, selective logging occurred in portions of the protected areas but logging last occurred in 1989 (*Aihara et al., 2016*). Although the protected areas are largely intact, the forest cover surrounding them has significantly declined due to intensive agricultural activities, particularly the establishment of oil palm plantations, and this land use change was ongoing at the time of the study (*Clements et al., 2010*; *Foo & Numata, 2019*).

## Population survey

Dung count-based surveys were conducted to CITES MIKE program standards (*Hedges & Lawson, 2006*). From late April to the end of August 2008, we used line transect methods to determine elephant dung-pile density (*Buckland et al., 2001*; *Hedges & Lawson, 2006*). The 1.5 km long transects were arranged in clusters along short baselines, with the clusters located systematically (but with a randomly-selected initial coordinate) across the 2,500 km$^2$ study area in order to give good geographical coverage. Each cluster had six transects unless part of it fell outside the study area (Fig. 2).

Estimating elephant density from the dung-pile density requires data on rates of elephant defecation and dung-pile decay. Following *Hedges & Lawson (2006)*, we used a mean defecation rate of 18.07 defecations per 24 h with standard error 0.0698; these data were derived from a study of free-ranging elephants in Indonesia (*Hedges et al., 2005*). We calculated dung decay rate using the method of *Laing et al. (2003)*, which entailed locating cohorts of fresh dung-piles prior to the line transect survey and then revisiting the marked dung-piles half-way through the overall line transect survey period to establish whether they were still present or had decayed. We used logistic regression in program R (*R-Development-Core-Team, 2008*) to characterize the probability of decay as a function of time and estimated the mean time to decay from this function. We analyzed transect data using the program DISTANCE (*Thomas et al., 2010*).

The work was carried out in ERL with the permission of the Malaysian Government's Department of Wildlife and National Parks (DWNP) and the Johor National Parks Corporation (JNPC). Permission from an Institutional Animal Care and Use Committee (IACUC) or equivalent animal ethics committee was not necessary as only indirect methods of assessing elephant population status were used (counts of dung-piles along transects).

## Population viability analysis

To assess relative extinction risks for the ERL elephant population under different management scenarios, we used our survey data together with data from other populations of wild Asian elephants in order to conduct a population viability analysis (PVA) (*Beissinger & McCullough, 2002*; *Beissinger & Westphal, 1998*; *Boyce, 1992*). We built our models in VORTEX Version 9.99, an individual-based simulation program (*Lacy, Borbat & Pollak, 2005*; *Miller & Lacy, 2005*), which has been used for a number of population viability analyses for Asian elephants (*Armbruster, Fernando & Lande, 1999*; *Leimgruber et al., 2008*; *Tilson et al., 1994*).

*Tilson et al. (1994)* summarized expert opinion for their models of wild elephant population viability in Sumatra. Following *Leimgruber et al. (2008)*, we also drew on this source and *Sukumar (2003)* for our models (Tables 1 and 2). We calculated the elephant carrying capacity of the ERL based on its area (2,500 km$^2$) and *Sukumar*'s (*2003*) estimate that rainforests can support 0.1 elephants/km$^2$. No trend in carrying capacity was included in our models in order to avoid exaggerating extinction risk given that our primary concern is to model the impact of translocations and other forms of removal (poaching including snaring and retaliatory killing for crop raiding) over a relatively short period. Poaching is not included as a separate named threat in our models because its

**Table 1  Terms used in Figs. 3–8, Tables 5–7, and Table S1.**

| Abbreviations used in scenario names and figure legends | |
| --- | --- |
| FB | Female breeding rate (%) |
| BaseMort | Baseline mortality rates (Table 1) |
| No removal, very low removal, etc. | Elephant removal rate for translocation (see Table 2) |
| Mort20%lower | Baseline mortality rates reduced by 20% (Table 3) |
| Mort20%higher | Baseline mortality rates increased by 20% (Table 3) |
| 0C and 2C | No and 2 types of catastrophe (flood and disease), respectively |
| NoQ, Q30, Q50 | No quasi-extinction and quasi-extinction at 30 and 50 individuals, respectively |
| **Column-head abbreviations** | |
| det-r | the mean population deterministic growth rate, $r$ |
| stoc-r | the mean population stochastic growth rate, $r$ |
| SD(r) | standard deviation of the stochastic growth rate |
| PE | the final probability of population extinction |
| N-ext | the mean final population size for those iterations that do not become extinct |
| SD(n-ext) | the standard deviation for the mean final population size for those iterations that do not become extinct |
| N-all | the mean final population size for all populations, including those that went extinct (e.g., had a final size of 0) |
| SD(N-all) | the standard deviation for N-all |
| MedianTE | If at least 50% of the iterations went extinct, the median time to extinction in years; |
| MeanTE | Of those iterations that experience extinctions, the mean time to first population extinction (in years) |

effects can be simulated by simply treating the translocation-related removals as deaths due to poaching (the underlying model structure and thus the results being the same). As far as we know, no elephants were killed illegally for this in the ERL population during our study, although a small number have been subsequently been killed illegally. In addition, we adopted the assumption of *Tilson et al. (1994)* and *Sukumar (2003)* that male mortality rates for Asian elephants are higher than those of females because of selective poaching for ivory, competition for mates including fights with other males, and the higher metabolic demands resulting from *musth* and larger body size. The effects of such differential mortality rates are reflected in the female-biased sex ratios seen in wild elephant populations. Inter-calving interval has been reported as 4.5–5 years in southern India but *c.* 6 years in Indonesia (*Tilson et al., 1994*), so we assumed female reproductive rate was 0.18 offspring/mature female/year but also considered rates of 0.16 offspring/mature female/year and 0.20 offspring/mature female/year to be plausible and incorporated them in our sensitivity analyses. We assumed stochastic variation in environmental conditions equally affected reproduction and mortality and this variation was about 20% of the mean value (*Leimgruber et al., 2008*; *Tilson et al., 1994*). We modeled the ERL population as a single closed population, with no migration to or from other areas in Peninsular

**Table 2 Baseline parameter values used for modeling the Endau Rompin Landscape elephant population.**

| Input parameter | Value | Source/justification |
|---|---|---|
| *General parameters* | | |
| Number of years | 100 | Following *Tilson et al. (1994)*; also see 'Discussion' section. |
| Time-steps | 1 year | Following *Tilson et al. (1994)* and *Leimgruber et al. (2008)*. |
| Number of iterations | 500 | Following *Tilson et al. (1994)*; 500–1,000 iterations are typical values in VORTEX models (*Miller & Lacy (2005)*). |
| Extinction definition | Only 1 sex remains | Following *Tilson et al. (1994)* and *Leimgruber et al. (2008)*, this is the standard definition of extinction in PVA analyses; two levels of quasi-extinction were also modeled, see text for further discussion. |
| *Reproductive systems (polygynous)* | | |
| Age of first offspring for females (years) | 20 | Following *Tilson et al. (1994)* who argue that females tend to breed later in rainforest areas compared to the more open areas of southern India. |
| Age of first offspring for males (years) | 20 | Following *Tilson et al. (1994)*. |
| Maximum age of reproduction (years) | 60 | Following *Tilson et al. (1994)*, *Sukumar (2003)*, and *Leimgruber et al. (2008)*. |
| Maximum number of progeny per year | 1 | Following *Tilson et al. (1994)*, *Sukumar (2003)*, and *Leimgruber et al. (2008)*. |
| Sex ratio at birth | 1:1 | Following *Tilson et al. (1994)* and *Leimgruber et al. (2008)*. |
| Density-dependent reproduction | No | Following *Tilson et al. (1994)* and *Leimgruber et al. (2008)*. |
| *Reproductive rates* | | |
| offspring/mature female/year | 0.18 | Following *Tilson et al. (1994)* and *Leimgruber et al. (2008)*. |
| Environmental variation in breeding | 3.20% | Approximately 20% of the mean value following *Tilson et al. (1994)* and *Leimgruber et al. (2008)*. |
| *Mortality rates for females* | | |
| 0–1 years | 15.00% | Following *Tilson et al. (1994)*, *Sukumar (2003)*, and *Leimgruber et al. (2008)*. |
| >1–5 | 4.00% | Following *Tilson et al. (1994)* and *Leimgruber et al. (2008)*. |
| >5–15 | 2.00% | Following *Tilson et al. (1994)* and *Leimgruber et al. (2008)*. |
| >15 | 2.50% | Following *Tilson et al. (1994)* and *Leimgruber et al. (2008)*. |
| *Mortality rates for males* | | |
| 0–1 | 15.00% | Following *Tilson et al. (1994)*, *Sukumar (2003)*, and *Leimgruber et al. (2008)*. |
| >1–5 | 5.00% | Following *Tilson et al. (1994)* and *Leimgruber et al. (2008)*. |
| >5–15 | 3.00% | Following *Sukumar (2003)* and *Leimgruber et al. (2008)*. |
| >15 | 3.00% | Following *Sukumar (2003)* and *Leimgruber et al. (2008)*. |
| *Mate monopolization* | | |
| Percent males in breeding pool | 80% | Following *Tilson et al. (1994)* and *Leimgruber et al. (2008)*. |
| *Initial population* | | |
| Start with age distribution | Stable | Following *Tilson et al. (1994)* and *Leimgruber et al. (2008)*; also see Table 3 |
| Initial population size | 135 | This study. |

**Table 2** (*continued*)

| Input parameter | Value | Source/justification |
|---|---|---|
| *Carrying capacity* | | |
| Carrying capacity (K) | 250 | Calculate from area of ERL using 0.1 elephant/sq km after *Sukumar (2003).* |
| SD in K due to environmental variation | 5 | Following *Leimgruber et al. (2008)*. |
| Trend in K? | No | Following *Leimgruber et al. (2008)* and most of the *Tilson et al. (1994)* scenarios; see text for further justification. |
| *Inbreeding depression* | | |
| Lethal equivalents | 3.14 | Following *Tilson et al. (1994)* and *Miller & Lacy (2005)*; the value is the mean for 40 mammalian species. |
| Percent due to recessive lethals | 50 | Following *Tilson et al. (1994)* and *Miller & Lacy (2005)*; the value is the mean for 40 mammalian species. |

Malaysia, based on recent survey and habitat connectivity data (*Gumal et al., 2009*) as well as the authors' personal observations and local DWNP staff's observations (S Saaban, pers. comm., 2007). We kept the basic parameter values shown in Table 2 constant in all models. Each model was run over 100 years with 1-year time steps and 500 iterations.

We considered five levels of elephant removal (permanent translocation out of the ERL), these ranged from no removal to a high rate of six animals per year (Table 3). These rates, especially the 'very low' and 'low' rates, are considered realistic based on the history of translocation in the ERL area. The removal scenarios of Table 3 also reflect the typical intention of the DWNP capture teams to translocate family units (*DWNP, 2006*). We modeled scenarios with and without catastrophes, which were defined as floods and disease. Following *Tilson et al. (1994)*, a 4% probability of drought lowering fertility by 40% and killing 5% of individuals, and a 1% probability of disease killing 10% of individuals was assumed.

The ERL elephant population was considered extinct if one of the sexes declined to zero but we also included two levels of quasi-extinction, defined as population size declining below 30 and 50 individuals, respectively. To determine the robustness of our baseline models, we conducted a sensitivity analysis. Following *Leimgruber et al. (2008)*, we increased and decreased the most important vital rates (number of offspring per mature female per year and mortality rate) as discussed above and shown in Table 4 and Table S1.

## RESULTS

### Population survey
#### Dung decay rate estimation
A total of 492 fresh dung-piles were found in three large zones (Rompin, Selai, and Peta) spread across the study area, monitored from 27 August 2007 to 30 May 2008, and classified during the second and third weeks of June 2008. Of those 492 dung-piles, 48 were not found again or were destroyed by construction works. The data for the remaining 446 dung-piles were used in the analyses. Logistic regression indicated a mean time to disappear of 308.67 days (SE = 16.01), which is within the expected range for Southeast Asian rain forests (*Hedges et al., 2005*).

**Table 3  Elephant removal rates included in the population viability models.**

| Scenario | Frequency | Total number of elephants removed | Adult females (≥20 yrs old) | Juvenile females (≥5 but <20 yrs old) | Adult males (≥20 yrs old) | Juvenile males (≥5 but <20 yrs old) |
|---|---|---|---|---|---|---|
| No removal | N/A | 0 | 0 | 0 | 0 | 0 |
| Very low removal | Every other year | 3 | 2 | 0 | 1 | 0 |
| Low removal | Every year | 3 | 2 | 0 | 1 | 0 |
| Medium removal | Every other year | 10 | 4 | 2 | 2 | 2 |
| High removal | Every year | 6 | 3 | 1 | 1 | 1 |

**Table 4  Male and female mortality rates used in the sensitivity analyses that were run to assess the robustness of the baseline models; three values for female breeding rate were also used in these analyses: 0.16, 0.18, and 0.20 offspring/mature female/y.**

| Age class (years) | Female mortality (%) | | | Male mortality (%) | | |
|---|---|---|---|---|---|---|
| | Baseline rates | Baseline rates reduced by 20% | Baseline rates increased by 20% | Baseline rates | Baseline rates reduced by 20% | Baseline rates increased by 20% |
| 0–1 (3f; 3m) | 15.00% | 12.00% | 18.00% | 15.00% | 12.00% | 18.00% |
| >1–5 (9f; 9m) | 4.00% | 3.20% | 4.80% | 5.00% | 4.00% | 6.00% |
| >5–15 (19f; 17m) | 2.00% | 1.60% | 2.40% | 3.00% | 2.40% | 3.60% |
| >15 (43f; 32m) | 2.50% | 2.00% | 3.00% | 3.00% | 2.40% | 3.60% |

*Line transect-based survey*

During the 4-month (late April–late August 2008) line transect-based survey, we found 226 elephant dung-piles along line transects totaling 194.56 km in length. Applying a mean defecation rate of 18.07 (SE = 0.0698) dung-piles per 24-hours and the decay rate given above, we estimated population density as 0.0538 (95% CI [0.0322–0.0901]) elephants/km$^2$ and population size as 135 (95% CI [80–225]) elephants in the 2,500 km$^2$ study area.

## Population viability analysis

A total of 234 scenarios were analyzed (Tables 5–7; Figs. 3–8; Table S1). The results suggest that the ERL elephant population could be self-sustaining provided no animals are removed for translocation or killed (and the basic assumptions of the PVA model are met). Our baseline scenarios gave a growth rate of $r = 0.006$ in the absence of catastrophes (flood and disease) and $r = 0.004$ when we included catastrophes in the models. All baseline scenarios returned a 0% probability of extinction in the absence of removals (Table 5; Fig. 3). Reducing the natality rate from 0.18 to 0.16 offspring/mature female/year, a rate also considered to be realistic based on data from Indonesia, results in growth rates of $r = 0$ and 0.003 with and without catastrophes, respectively, but still returns a 0% probability of extinction in the absence of removals (Table 6; Fig. 4). Under the most optimistic scenarios (natality rate of 0.20 offspring/mature female/year, mortality rates reduced by 20%), the ERL population has a 0% probability of extinction and grows at a rate of $r = 0.013$ and 0.015 with and without catastrophes, respectively (Table 7; Fig. 5).

Saaban et al. (2020), *PeerJ*, DOI 10.7717/peerj.8209

**Table 5   Results of the population viability analysis for all baseline scenarios.** See Table 1 for terms used.

| Scenario name | det-r | stoc-r | SD(r) | PE | N-ext | SD (N-ext) | N-all | SD (N-all) | MedianTE | MeanTE |
|---|---|---|---|---|---|---|---|---|---|---|
| FB18% + BaseMort + 0C + no removal + NoQ | 0.006 | 0.006 | 0.025 | 0.000 | 218.24 | 28.94 | 218.24 | 28.94 | 0 | 0.0 |
| FB18% + BaseMort + 0C + no removal + Q30 | 0.006 | 0.005 | 0.025 | 0.000 | 216.57 | 32.39 | 216.57 | 32.39 | 0 | 0.0 |
| FB18% + BaseMort + 0C + no removal + Q50 | 0.006 | 0.006 | 0.025 | 0.000 | 220.48 | 28.04 | 220.48 | 28.04 | 0 | 0.0 |
| FB18% + BaseMort + 2C + no removal + NoQ | 0.004 | 0.003 | 0.03 | 0.000 | 186.70 | 42.40 | 186.70 | 42.40 | 0 | 0.0 |
| FB18% + BaseMort + 2C + no removal + Q30 | 0.004 | 0.003 | 0.030 | 0.000 | 186.24 | 42.12 | 186.24 | 42.12 | 0 | 0.0 |
| FB18% + BaseMort + 2C + no removal + Q50 | 0.004 | 0.003 | 0.030 | 0.000 | 186.65 | 43.28 | 186.65 | 43.28 | 0 | 0.0 |
| FB18% + BaseMort + 0C + very low removal + NoQ | 0.006 | −0.032 | 0.067 | 0.638 | 27.24 | 25.53 | 10.27 | 20.02 | 93 | 85.4 |
| FB18% + BaseMort + 0C + very low removal + Q30 | 0.006 | −0.019 | 0.039 | 0.906 | 58.30 | 23.21 | 8.88 | 18.43 | 75 | 73.0 |
| FB18% + BaseMort + 0C + very low removal + Q50 | 0.006 | −0.015 | 0.034 | 0.932 | 66.62 | 12.76 | 9.01 | 18.08 | 63 | 63.2 |
| FB18% + BaseMort + 2C + very low removal + NoQ | 0.004 | −0.039 | 0.076 | 0.804 | 20.72 | 18.20 | 4.35 | 11.46 | 85 | 81.0 |
| FB18% + BaseMort + 2C + very low removal + Q30 | 0.004 | −0.022 | 0.042 | 0.972 | 46.50 | 13.84 | 2.98 | 8.82 | 65 | 66.0 |
| FB18% + BaseMort + 2C + very low removal + Q50 | 0.004 | −0.017 | 0.037 | 0.976 | 61.08 | 12.29 | 3.93 | 11.61 | 55 | 56.6 |
| FB18% + BaseMort + 0C + low removal + NoQ | 0.006 | −0.078 | 0.087 | 1.000 | 0.00 | 0.00 | 0.00 | 0.00 | 46 | 46.5 |
| FB18% + BaseMort + 0C + low removal + Q30 | 0.006 | −0.046 | 0.037 | 1.000 | 0.00 | 0.00 | 0.00 | 0.00 | 33 | 33.0 |
| FB18% + BaseMort + 0C + low removal + Q50 | 0.006 | −0.037 | 0.031 | 1.000 | 0.00 | 0.00 | 0.00 | 0.00 | 27 | 27.0 |
| FB18% + BaseMort + 2C + low removal + NoQ | 0.004 | −0.082 | 0.09 | 1.000 | 0.00 | 0.00 | 0.00 | 0.00 | 45 | 44.6 |
| FB18% + BaseMort + 2C + low removal + Q30 | 0.004 | −0.048 | 0.04 | 1.000 | 0.00 | 0.00 | 0.00 | 0.00 | 31 | 31.5 |
| FB18% + BaseMort + 2C + low removal + Q50 | 0.004 | −0.038 | 0.034 | 1.000 | 0.00 | 0.00 | 0.00 | 0.00 | 25 | 25.9 |
| FB18% + BaseMort + 0C + medium removal + NoQ | 0.006 | −0.097 | 0.138 | 1.000 | 0.00 | 0.00 | 0.00 | 0.00 | 37 | 37.7 |
| FB18% + BaseMort + 0C + medium removal + Q30 | 0.006 | −0.058 | 0.07 | 1.000 | 0.00 | 0.00 | 0.00 | 0.00 | 25 | 25.3 |
| FB18% + BaseMort + 0C + medium removal + Q50 | 0.006 | −0.048 | 0.059 | 1.000 | 0.00 | 0.00 | 0.00 | 0.00 | 20 | 20.2 |
| FB18% + BaseMort + 2C + medium removal + NoQ | 0.004 | −0.099 | 0.137 | 1.000 | 0.00 | 0.00 | 0.00 | 0.00 | 37 | 36.8 |
| FB18% + BaseMort + 2C + medium removal + Q30 | 0.004 | −0.061 | 0.073 | 1.000 | 0.00 | 0.00 | 0.00 | 0.00 | 25 | 24.2 |
| FB18% + BaseMort + 2C + medium removal + Q50 | 0.004 | −0.05 | 0.06 | 1.000 | 0.00 | 0.00 | 0.00 | 0.00 | 19 | 19.3 |
| FB18% + BaseMort + 0C + high removal + NoQ | 0.006 | −0.105 | 0.073 | 1.000 | 0.00 | 0.00 | 0.00 | 0.00 | 28 | 28.1 |
| FB18% + BaseMort + 0C + high removal + Q30 | 0.006 | −0.08 | 0.044 | 1.000 | 0.00 | 0.00 | 0.00 | 0.00 | 19 | 19.1 |
| FB18% + BaseMort + 0C + high removal + Q50 | 0.006 | −0.067 | 0.036 | 1.000 | 0.00 | 0.00 | 0.00 | 0.00 | 15 | 15.1 |
| FB18% + BaseMort + 2C + high removal + NoQ | 0.004 | −0.111 | 0.082 | 1.000 | 0.00 | 0.00 | 0.00 | 0.00 | 28 | 27.9 |
| FB18% + BaseMort + 2C + high removal + Q30 | 0.004 | −0.082 | 0.046 | 1.000 | 0.00 | 0.00 | 0.00 | 0.00 | 19 | 18.6 |
| FB18% + BaseMort + 2C + high removal + Q50 | 0.004 | −0.068 | 0.038 | 1.000 | 0.00 | 0.00 | 0.00 | 0.00 | 15 | 14.7 |

Saaban et al. (2020), *PeerJ*, DOI 10.7717/peerj.8209

**Table 6  Results of the population viability analysis for all reduced female breeding rate scenarios (0.16 offspring/mature female/year, all other parameter values the same as in the baseline scenarios).** See Table 1 for terms used.

| Scenario name | det-r | stoc-r | SD(r) | PE | N-ext | SD (N-ext) | N-all | SD (N-all) | MedianTE | MeanTE |
|---|---|---|---|---|---|---|---|---|---|---|
| FB16% + BaseMort + 0C + no removal + NoQ | 0.003 | 0.002 | 0.025 | 0.000 | 174.02 | 38.02 | 174.02 | 38.02 | 0 | 0.0 |
| FB16% + BaseMort + 0C + no removal + Q30 | 0.003 | 0.002 | 0.025 | 0.000 | 172.47 | 38.53 | 172.47 | 38.53 | 0 | 0.0 |
| FB16% + BaseMort + 0C + no removal + Q50 | 0.003 | 0.002 | 0.025 | 0.000 | 175.00 | 38.33 | 175.00 | 38.33 | 0 | 0.0 |
| FB16% + BaseMort + 2C + no removal + NoQ | 0.000 | 0.000 | 0.031 | 0.000 | 139.21 | 38.83 | 139.21 | 38.83 | 0 | 0.0 |
| FB16% + BaseMort + 2C + no removal + Q30 | 0.000 | 0.000 | 0.030 | 0.000 | 136.88 | 40.24 | 136.88 | 40.24 | 0 | 0.0 |
| FB16% + BaseMort + 2C + no removal + Q50 | 0.000 | 0.000 | 0.030 | 0.002 | 144.00 | 39.27 | 143.79 | 39.52 | 0 | 71.0 |
| FB16% + BaseMort + 0C + very low removal + NoQ | 0.003 | −0.041 | 0.076 | 0.852 | 12.38 | 11.33 | 2.07 | 6.19 | 83 | 81.2 |
| FB16% + BaseMort + 0C + very low removal + Q30 | 0.003 | −0.022 | 0.040 | 0.984 | 54.25 | 27.61 | 2.07 | 8.21 | 64 | 65.6 |
| FB16% + BaseMort + 0C + very low removal + Q50 | 0.003 | −0.017 | 0.034 | 0.998 | 91.00 | 0.00 | 2.03 | 6.85 | 55 | 55.9 |
| FB16% + BaseMort + 2C + very low removal + NoQ | 0.000 | −0.045 | 0.081 | 0.948 | 10.23 | 7.98 | 0.63 | 2.90 | 77 | 77.1 |
| FB16% + BaseMort + 2C + very low removal + Q30 | 0.000 | −0.025 | 0.042 | 0.994 | 44.00 | 10.58 | 0.92 | 4.26 | 59 | 60.1 |
| FB16% + BaseMort + 2C + very low removal + Q50 | 0.000 | −0.020 | 0.037 | 1.000 | 0.00 | 0.00 | 0.71 | 2.98 | 49 | 49.4 |
| FB16% + BaseMort + 0C + low removal + NoQ | 0.003 | −0.082 | 0.088 | 1.000 | 0.00 | 0.00 | 0.00 | 0.00 | 44 | 44.4 |
| FB16% + BaseMort + 0C + low removal + Q30 | 0.003 | −0.048 | 0.037 | 1.000 | 0.00 | 0.00 | 0.00 | 0.00 | 31 | 31.4 |
| FB16% + BaseMort + 0C + low removal + Q50 | 0.003 | −0.038 | 0.031 | 1.000 | 0.00 | 0.00 | 0.00 | 0.00 | 26 | 26.0 |
| FB16% + BaseMort + 2C + low removal + NoQ | 0.000 | −0.086 | 0.093 | 1.000 | 0.00 | 0.00 | 0.00 | 0.00 | 43 | 42.8 |
| FB16% + BaseMort + 2C + low removal + Q30 | 0.000 | −0.050 | 0.040 | 1.000 | 0.00 | 0.00 | 0.00 | 0.00 | 30 | 30.0 |
| FB16% + BaseMort + 2C + low removal + Q50 | 0.000 | −0.041 | 0.034 | 1.000 | 0.00 | 0.00 | 0.00 | 0.00 | 25 | 24.5 |
| FB16% + BaseMort + 0C + medium removal + NoQ | 0.003 | −0.100 | 0.138 | 1.000 | 0.00 | 0.00 | 0.00 | 0.00 | 37 | 36.5 |
| FB16% + BaseMort + 0C + medium removal + Q30 | 0.003 | −0.060 | 0.071 | 1.000 | 0.00 | 0.00 | 0.00 | 0.00 | 25 | 24.3 |
| FB16% + BaseMort + 0C + medium removal + Q50 | 0.003 | −0.050 | 0.059 | 1.000 | 0.00 | 0.00 | 0.00 | 0.00 | 19 | 19.5 |
| FB16% + BaseMort + 2C + medium removal + NoQ | 0.000 | −0.102 | 0.140 | 1.000 | 0.00 | 0.00 | 0.00 | 0.00 | 36 | 36.0 |
| FB16% + BaseMort + 2C + medium removal + Q30 | 0.000 | −0.063 | 0.073 | 1.000 | 0.00 | 0.00 | 0.00 | 0.00 | 23 | 23.4 |
| FB16% + BaseMort + 2C + medium removal + Q50 | 0.000 | −0.053 | 0.061 | 1.000 | 0.00 | 0.00 | 0.00 | 0.00 | 19 | 18.4 |
| FB16% + BaseMort + 0C + high removal + NoQ | 0.003 | −0.110 | 0.081 | 1.000 | 0.00 | 0.00 | 0.00 | 0.00 | 28 | 28.1 |
| FB16% + BaseMort + 0C + high removal + Q30 | 0.003 | −0.082 | 0.045 | 1.000 | 0.00 | 0.00 | 0.00 | 0.00 | 19 | 18.5 |
| FB16% + BaseMort + 0C + high removal + Q50 | 0.003 | −0.069 | 0.035 | 1.000 | 0.00 | 0.00 | 0.00 | 0.00 | 15 | 14.6 |
| FB16% + BaseMort + 2C + high removal + NoQ | 0.000 | −0.112 | 0.082 | 1.000 | 0.00 | 0.00 | 0.00 | 0.00 | 28 | 27.4 |
| FB16% + BaseMort + 2C + high removal + Q30 | 0.000 | −0.084 | 0.047 | 1.000 | 0.00 | 0.00 | 0.00 | 0.00 | 18 | 18.2 |
| FB16% + BaseMort + 2C + high removal + Q50 | 0.000 | −0.071 | 0.038 | 1.000 | 0.00 | 0.00 | 0.00 | 0.00 | 14 | 14.3 |

Saaban et al. (2020), *PeerJ*, DOI 10.7717/peerj.8209

**Table 7  Results of the population viability analysis for the most optimistic scenarios (0.20 offspring/mature female/year, mortality rates reduced by 20%, all other parameter values the same as in the baseline scenarios).** See Table 1 for terms used.

| Scenario name | det-r | stoc-r | SD (r) | PE | N-ext | SD (N-ext) | N-all | SD (N-all) | Median TE | Mean TE |
|---|---|---|---|---|---|---|---|---|---|---|
| FB20% + Mort20 %lower + 0C + no removal + NoQ | 0.015 | 0.014 | 0.022 | 0.000 | 244.43 | 5.80 | 244.43 | 5.80 | 0 | 0.0 |
| FB20% + Mort20 %lower + 0C + no removal + Q30 | 0.015 | 0.015 | 0.022 | 0.000 | 244.68 | 5.88 | 244.68 | 5.88 | 0 | 0.0 |
| FB20% + Mort20 %lower + 0C + no removal + Q50 | 0.015 | 0.015 | 0.022 | 0.000 | 244.71 | 5.53 | 244.71 | 5.53 | 0 | 0.0 |
| FB20% + Mort20 %lower + 2C + no removal + NoQ | 0.013 | 0.012 | 0.027 | 0.000 | 241.42 | 10.73 | 241.42 | 10.73 | 0 | 0.0 |
| FB20% + Mort20 %lower + 2C + no removal + Q30 | 0.013 | 0.012 | 0.027 | 0.000 | 242.10 | 9.31 | 242.10 | 9.31 | 0 | 0.0 |
| FB20% + Mort20 %lower + 2C + no removal + Q50 | 0.013 | 0.012 | 0.027 | 0.000 | 242.13 | 8.66 | 242.13 | 8.66 | 0 | 0.0 |
| FB20% + Mort20 %lower + 0C + very low removal + NoQ | 0.015 | −0.002 | 0.031 | 0.028 | 137.99 | 66.06 | 134.15 | 68.96 | 0 | 87.1 |
| FB20% + Mort20 %lower + 0C + very low removal + Q30 | 0.015 | −0.002 | 0.029 | 0.104 | 139.55 | 60.88 | 126.76 | 68.87 | 0 | 90.4 |
| FB20% + Mort20 %lower + 0C + very low removal + Q50 | 0.015 | −0.001 | 0.028 | 0.116 | 147.13 | 57.15 | 132.86 | 66.87 | 0 | 83.8 |
| FB20% + Mort20 %lower + 2C + very low removal + NoQ | 0.013 | −0.011 | 0.044 | 0.146 | 90.35 | 61.38 | 77.34 | 64.88 | 0 | 88.2 |
| FB20% + Mort20 %lower + 2C + very low removal + Q30 | 0.013 | −0.007 | 0.035 | 0.258 | 107.03 | 53.12 | 81.46 | 63.21 | 0 | 81.1 |
| FB20% + Mort20 %lower + 2C + very low removal + Q50 | 0.013 | −0.005 | 0.033 | 0.348 | 121.13 | 52.93 | 84.41 | 66.54 | 0 | 76.4 |
| FB20% + Mort20 %lower + 0C + low removal + NoQ | 0.015 | −0.062 | 0.075 | 1.000 | 0.00 | 0.00 | 0.00 | 0.00 | 52 | 52.6 |
| FB20% + Mort20 %lower + 0C + low removal + Q30 | 0.015 | −0.036 | 0.035 | 1.000 | 0.00 | 0.00 | 0.00 | 0.00 | 41 | 41.3 |
| FB20% + Mort20 %lower + 0C + low removal + Q50 | 0.015 | −0.028 | 0.030 | 1.000 | 0.00 | 0.00 | 0.00 | 0.00 | 35 | 35.3 |
| FB20% + Mort20 %lower + 2C + low removal + NoQ | 0.013 | −0.067 | 0.080 | 1.000 | 0.00 | 0.00 | 0.00 | 0.00 | 50 | 50.4 |
| FB20% + Mort20 %lower + 2C + low removal + Q30 | 0.013 | −0.039 | 0.039 | 1.000 | 0.00 | 0.00 | 0.00 | 0.00 | 38 | 38.3 |
| FB20% + Mort20 %lower + 2C + low removal + Q50 | 0.013 | −0.030 | 0.032 | 1.000 | 0.00 | 0.00 | 0.00 | 0.00 | 32 | 32.6 |
| FB20% + Mort20 %lower + 0C + medium removal + NoQ | 0.015 | −0.081 | 0.120 | 1.000 | 0.00 | 0.00 | 0.00 | 0.00 | 41 | 41.1 |
| FB20% + Mort20 %lower + 0C + medium removal + Q30 | 0.015 | −0.050 | 0.068 | 1.000 | 0.00 | 0.00 | 0.00 | 0.00 | 29 | 29.3 |
| FB20% + Mort20 %lower + 0C + medium removal + Q50 | 0.015 | −0.041 | 0.057 | 1.000 | 0.00 | 0.00 | 0.00 | 0.00 | 23 | 23.8 |
| FB20% + Mort20 %lower + 2C + medium removal + NoQ | 0.013 | −0.086 | 0.125 | 1.000 | 0.00 | 0.00 | 0.00 | 0.00 | 39 | 39.9 |
| FB20% + Mort20 %lower + 2C + medium removal + Q30 | 0.013 | −0.053 | 0.070 | 1.000 | 0.00 | 0.00 | 0.00 | 0.00 | 27 | 27.8 |
| FB20% + Mort20 %lower + 2C + medium removal + Q50 | 0.013 | −0.043 | 0.059 | 1.000 | 0.00 | 0.00 | 0.00 | 0.00 | 23 | 22.7 |

Saaban et al. (2020), *PeerJ*, DOI 10.7717/peerj.8209

**Table 7** (*continued*)

| Scenario name | det-r | stoc-r | SD (r) | PE | N-ext | SD (N-ext) | N-all | SD (N-all) | Median TE | Mean TE |
|---|---|---|---|---|---|---|---|---|---|---|
| FB20% + Mort20 %lower + 0C + high removal + NoQ | 0.015 | −0.087 | 0.060 | 1.000 | 0.00 | 0.00 | 0.00 | 0.00 | 30 | 29.7 |
| FB20% + Mort20 %lower + 0C + high removal + Q30 | 0.015 | −0.070 | 0.041 | 1.000 | 0.00 | 0.00 | 0.00 | 0.00 | 22 | 21.7 |
| FB20% + Mort20 %lower + 0C + high removal + Q50 | 0.015 | −0.060 | 0.035 | 1.000 | 0.00 | 0.00 | 0.00 | 0.00 | 17 | 16.7 |
| FB20% + Mort20 %lower + 2C + high removal + NoQ | 0.013 | −0.091 | 0.064 | 1.000 | 0.00 | 0.00 | 0.00 | 0.00 | 30 | 29.3 |
| FB20% + Mort20 %lower + 2C + high removal + Q30 | 0.013 | −0.073 | 0.043 | 1.000 | 0.00 | 0.00 | 0.00 | 0.00 | 21 | 20.9 |
| FB20% + Mort20 %lower + 2C + high removal + Q50 | 0.013 | −0.062 | 0.038 | 1.000 | 0.00 | 0.00 | 0.00 | 0.00 | 16 | 16.3 |

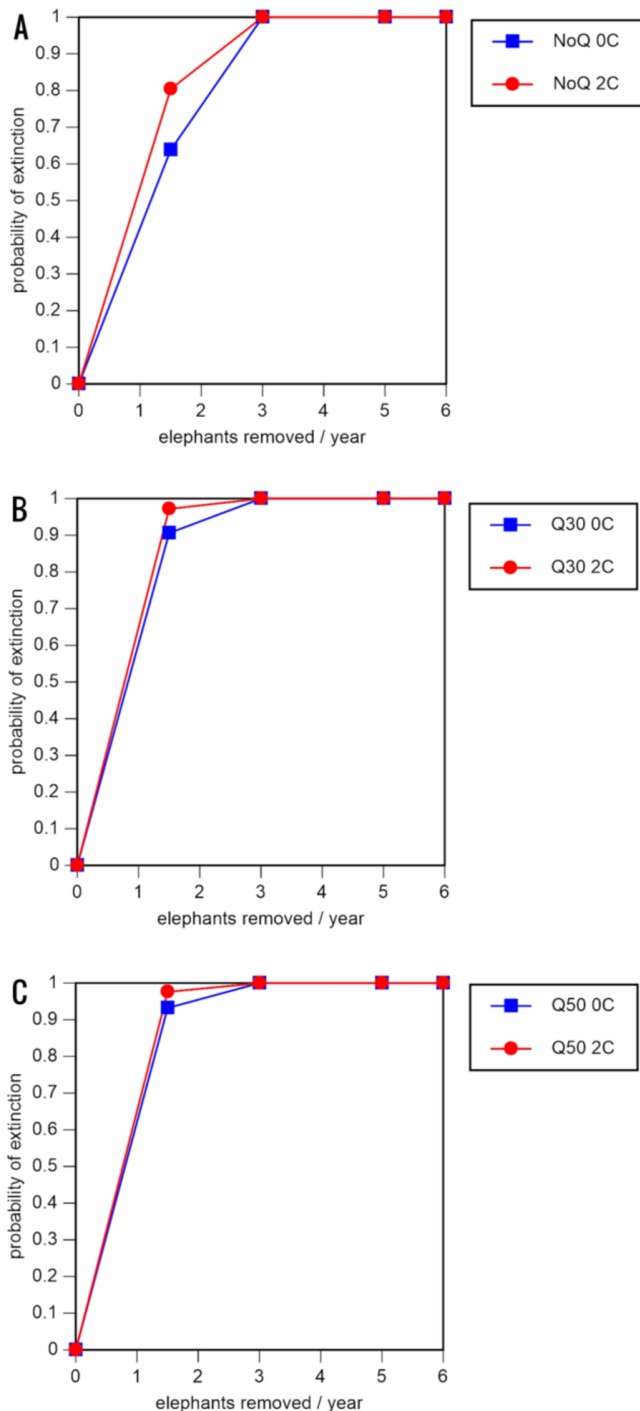

**Figure 3** Results of the population viability analysis for all baseline scenarios showing the effect of different elephant removal rates on (A) the probability of extinction, (B) the probability of quasi-extinction at 30 animals (shown as Q30), and (C) the probability of quasi-extinction at 50 animals (shown as Q50), with and without catastrophes, flood and disease (shown as 0C and 2C) For values see Table 4 and for terms used see Table 1.

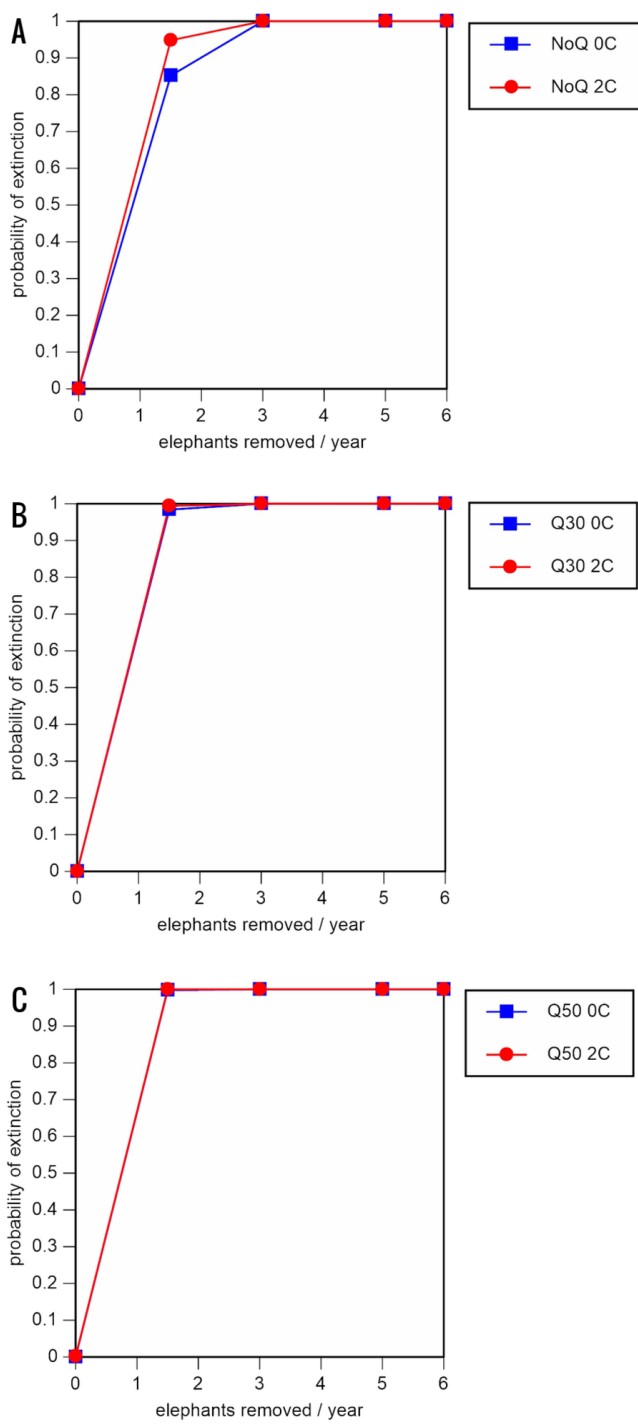

**Figure 4** Results of the population viability analysis for all reduced female breeding rate scenarios (natality rate of 0.16 offspring/mature female/year, all other parameter values the same as in the baseline scenarios) showing the effect of different elephant removal rates on (A) the probability of extinction, (B) the probability of quasi-extinction at 30 animals (shown as Q30), and (C) the probability of quasi-extinction at 50 animals (shown as Q50), with and without catastrophes, flood and disease (shown as 0C and 2C) For values see Table 5 and for terms used see Table 1.

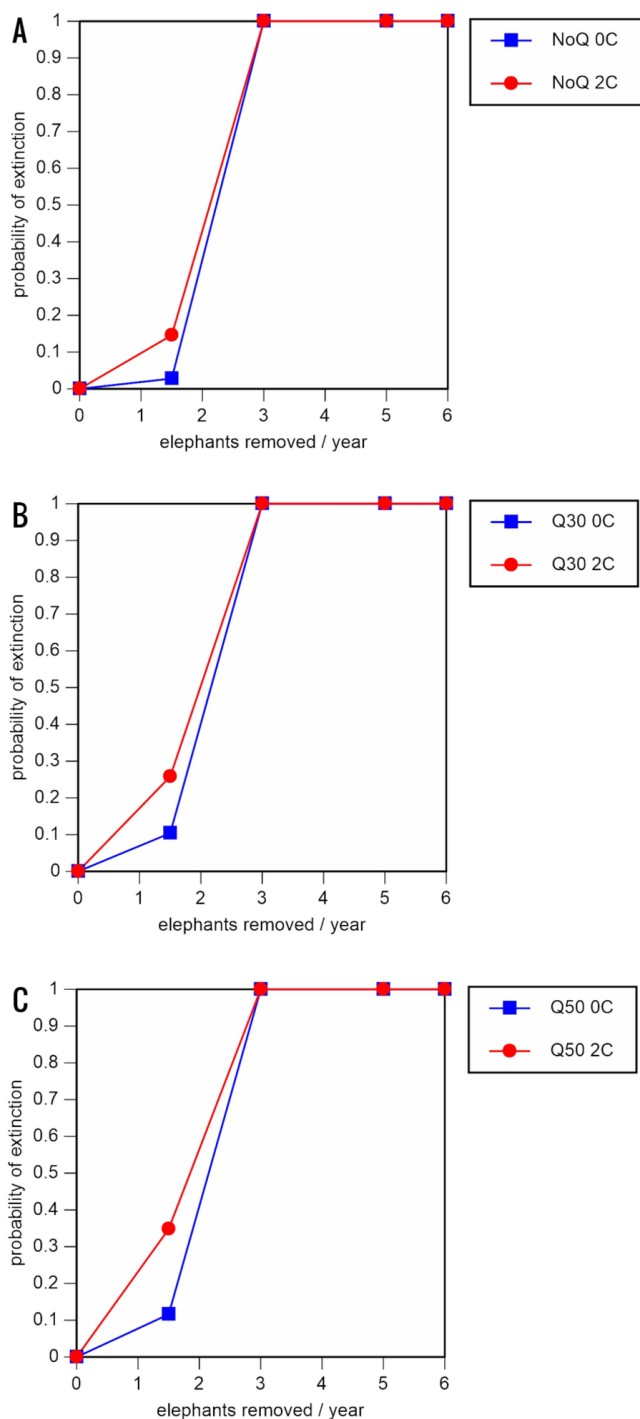

**Figure 5** Results of the population viability analysis for the most optimistic scenarios (natality rate of 0.20 offspring/mature female/year, mortality rates reduced by 20%, all other parameter values the same as in the baseline scenarios), showing the effect of different elephant removal rates on (A) the probability of extinction, (B) the probability of quasi-extinction at 30 animals (shown as Q30), and (C) the probability of quasi-extinction at 50 animals (shown as Q50), with and without catastrophes, flood and disease (shown as 0C and 2C) For values see Table 6 and for terms used see Table 1.

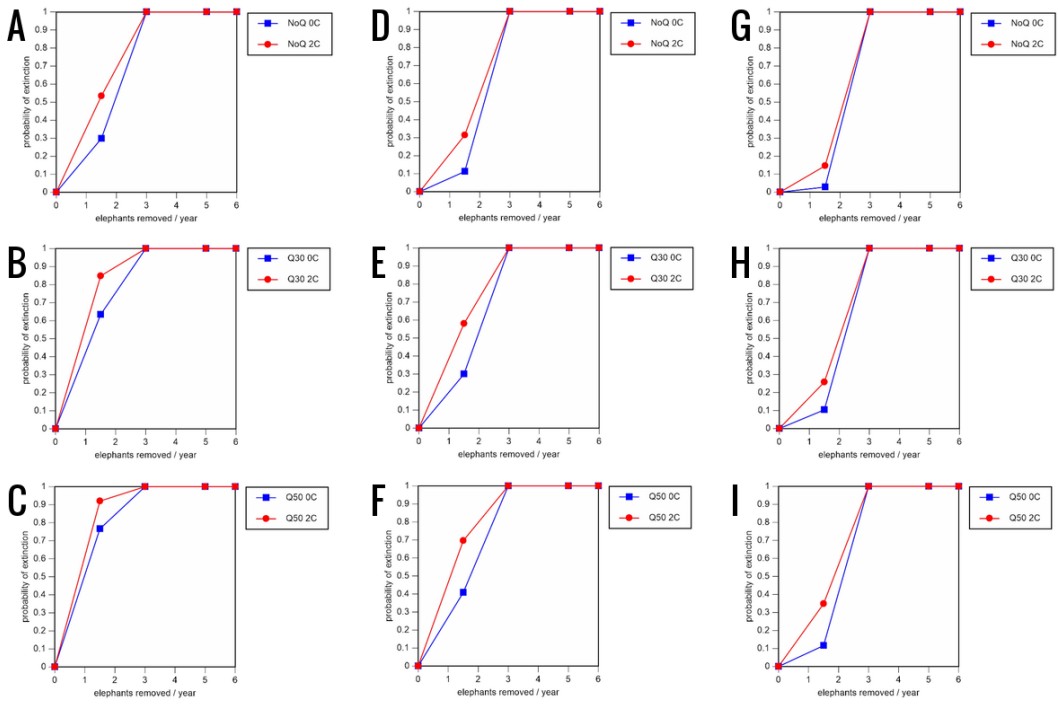

**Figure 6   Results of the sensitivity analysis for the PVA models with mortality rates reduced by 20% and three different natality rates.** 0.16 offspring/mature female/year (A –C), 0.18 offspring/mature female/year (D –F), and 0.20 offspring/mature female/year (G –I) (all other parameter values the same as in the baseline scenarios), showing the effect of different elephant removal rates on the probability of extinction (and quasi-extinction at 30 and 50 animals, shown as Q30 and Q50) with and without catastrophes (flood and disease, shown as 0C and 2C). For values see the Supplemental Information and for terms used see Table 1.

Including elephant removals in the models results in very high probabilities of extinction in all scenarios considered realistic. Those scenarios with very low removal rates (3 animals removed, every other year; Table 3) and no catastrophes have probabilities of extinction of 63.8–85.2% over a 100-year period, with mean times to extinction of 81.2–85.4 years (i.e., <3 elephant generations), while those scenarios with low removal rates (3 animals removed every year) have a 100% probability of extinction and a mean time to extinction of 44.4–46.5 years in the absence of catastrophes (Tables 5 & Table 6). Even the most optimistic scenarios return a 100% probability of extinction and a mean time to extinction of 52.6 years when low removal rates—but no catastrophes—are included in the models (Table 7). On the other hand, a high rate of capture (6 animals removed every year) is predicted to lead to the extinction of the ERL elephant population in *c.* 27–29 years if catastrophes are included in the models (mean time to extinction 27.4–27.9 years; Tables 5–7; Figs. 3–5).

All our models were robust, with changes in natality and mortality rates of up to 20% causing only minor changes in growth rates, probability of extinction, or mean time to extinction, and thus had no qualitative effects on our conclusions. Most notably, all the sensitivity analysis scenarios with the low capture rate (3 animals removed every year)

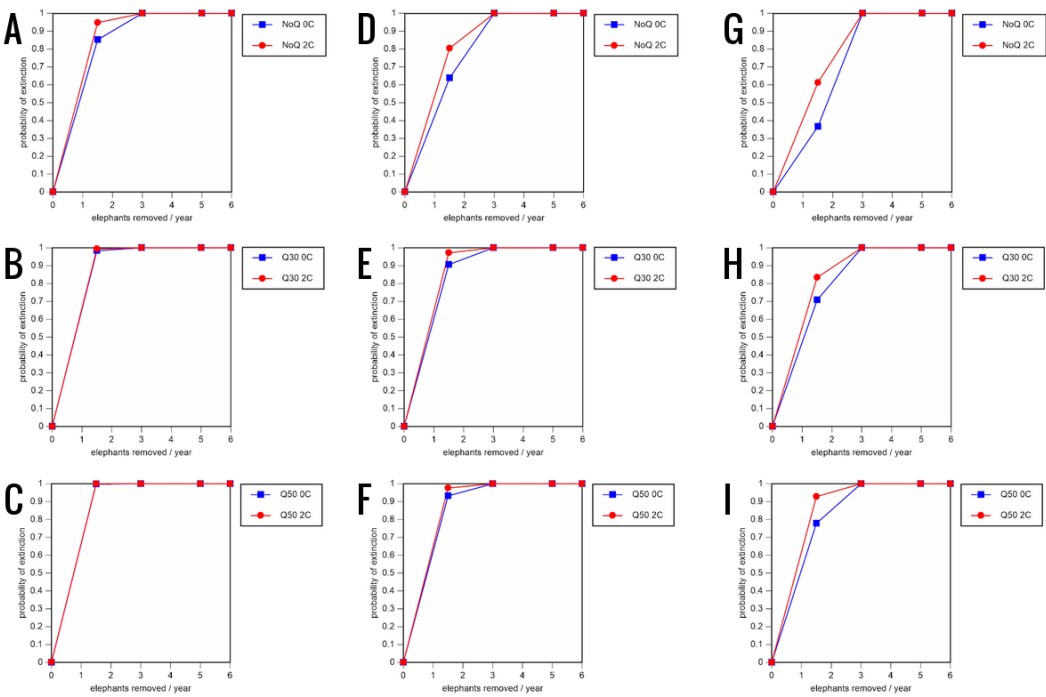

**Figure 7** **Results of the sensitivity analysis for the PVA models with baseline mortality rates and three different natality rates.** 0.16 offspring/mature female/year (A–C), 0.18 offspring/mature female/year (D–F), and 0.20 offspring/mature female/year (G–I) (all other parameter values the same as in the baseline scenarios), showing the effect of different elephant removal rates on the probability of extinction (and quasi-extinction at 30 and 50 animals, shown as Q30 and Q50) with and without catastrophes (flood and disease, shown as 0C and 2C). For values see the Supplemental Information and for terms used see Table 1.

resulted in a 100% probability of extinction regardless of other parameter values (Figs. 6–8; Supporting information).

## DISCUSSION

### The need for science-based conservation management

Species conservation is more effective when it is based on good science and reliable evidence but too often this is not the case (*Hayward et al., 2015*; *Pullin & Knight, 2001*; *Sutherland et al., 2004*). While there is a growing appreciation of the dangers of making interventions without a proper understanding of their impact or effectiveness, this appreciation is growing too slowly and is failing to have sufficient impact on conservation practice, even for high profile species such as elephants (*Elephas maximus*, *Loxodonta africana*) and tigers (*Panthera tigris*) (*Blake & Hedges, 2004*; *Hedges & Gunaryadi, 2009*; *Karanth et al., 2003*; *Young & Van Aarde, 2011*). Moreover, there is an increasingly recognized need for conservation scientists to produce research of greater relevance to conservation practitioners (*Laurance et al., 2012*), and to bridge the gap between research and publication on the one hand and implementation on the other (*Arlettaz et al., 2010*; *Meijaard & Sheil, 2007*; *Meijaard, Sheil & Cardillo, 2014*). This study provides an example of conservation

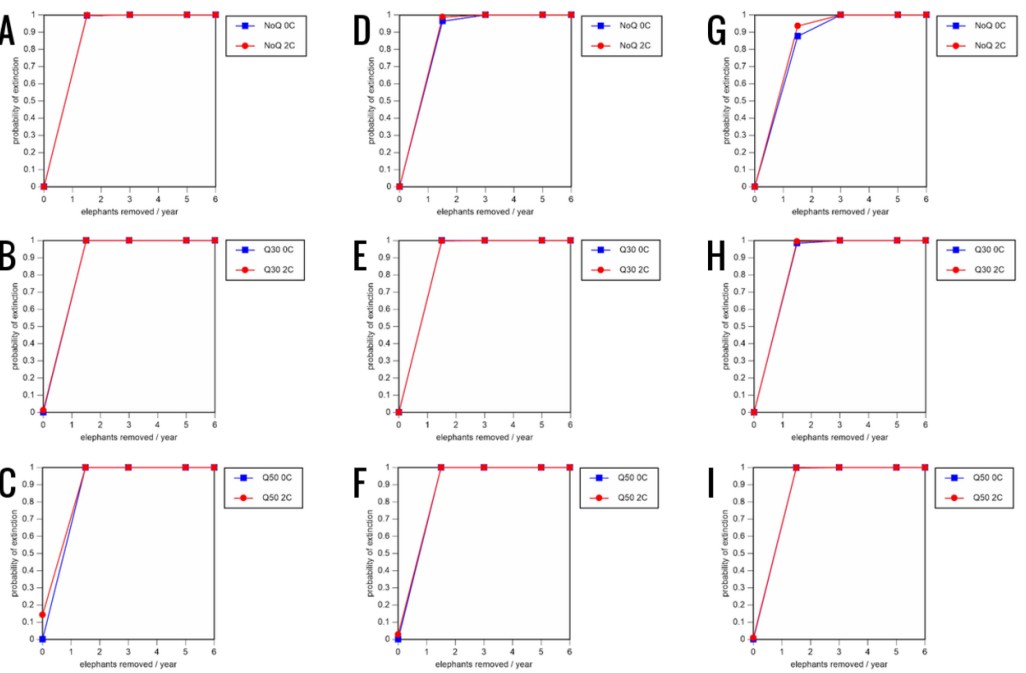

**Figure 8** Results of the sensitivity analysis for the PVA models with mortality rates increased by 20% and three different natality rates: 0.16 offspring/mature female/year (A–C), 0.18 offspring/mature female/year (D–F), and 0.20 offspring/mature female/year (G–I) (all other parameter values the same as in the baseline scenarios), showing the effect of different elephant removal rates on the probability of extinction (and quasi-extinction at 30 and 50 animals, shown as Q30 and Q50) with and without catastrophes (flood and disease, shown as 0C and 2C). For values see the Supplemental Information and for terms used see Table 1.

scientists working alongside practitioners and policy makers to address a question of immediate relevance to the conservation of wildlife, in this case how best to protect an important population of elephants, jointly publishing the results and—critically—using them to inform wildlife management policy and practice in Malaysia including the recent (2013) National Elephant Conservation Action Plan (NECAP) for Peninsular Malaysia (*DWNP, 2013*). Specifically, scientists from the Wildlife Conservation Society (WCS, an international NGO with a national program in Malaysia) worked alongside practitioners and policy makers from the Department of Wildlife and National Parks (DWNP) and the Johor National Parks Corporation (JNPC) to assess the size and viability of the ERL elephant population. The results of the study were then used by staff from DWNP, JNPC, and WCS to help prepare the National Elephant Conservation Action Plan for Peninsular Malaysia, which was published in 2013, after a series of workshops convened by DWNP and WCS over 2011–2013 and featuring inputs from the Department of Town and Country Planning (DTCP), NGOs, universities, and other representatives of civil society. In addition, staff from DWNP, JNPC, DTCP, and WCS are all authors of this paper.

## Significance of Endau Rompin's elephant population

The ERL elephant population estimate, 135 (95% CI [80–225]) elephants, is only the second such estimate for Peninsular Malaysia to be based on modern sampling-based methods (*Clements et al., 2010*), the first being the 2007 population estimate of 631 (95% CI [436–915]) elephants in Taman Negara, which also resulted from a DWNP/WCS project (*Hedges, Gumal & Ng, 2008*). The estimated population density of 0.0538 (95% CI [0.0322–0.0901]) elephants/km$^2$ in the ERL is somewhat lower than the 0.1 elephants/km$^2$ that *Sukumar (2003)* suggests Asian rainforests can support (although note the upper confidence limit) and considerably lower than the 0.57 elephants/km$^2$ reported by *Hedges et al. (2005)* for a rainforest area in nearby Sumatra. These lower densities may reflect differences in habitat quality but are perhaps more likely to be an indication of the effect of previous translocations of elephants out of the ERL as well as possible losses to poachers or retaliatory killing for HEC. Nevertheless, our results suggest that the elephant population in the ERL is of clear national importance and indeed regional importance given (1) the preponderance of small (<500) elephant populations in highly fragmented habitat in Southeast Asia (*Hedges, Fisher & Rose, 2009*; *Leimgruber et al., 2003*); (2) that, with effective protection, the population could at least double in size to the estimated carrying capacity of approximately 250 elephants (a doubling in elephant numbers would take *c.* 23–35 years if population annual growth rates could be increased to 2–3%); and (3) there is still an opportunity for gene flow to be re-established with other elephant populations within the Central Forest Spine (CFS) to the north since the Master Plan for the CFS envisages 51,000 km$^2$ of contiguous forests, with protected core areas, including those in the ERL, linked within the greater landscape by ecological corridors (*Brodie et al., 2016*; *DTCP, 2009*). However, the challenges of re-establishing connectivity for elephants should not be underestimated given the risk of further deforestation, the shortage of resources to implement the CFS, and governance conflicts between Federal and State governments (*Maniam & Singaravelloo, 2015*).

## Population viability analysis and the effects of translocations

Population viability modeling is sometimes controversial because the requisite data are often lacking. In order to minimize such difficulties, we followed the recommendations of *Beissinger & Westphal (1998)* and *Burgman & Possingham (2000)* in treating our results as relative, rather than absolute, estimates of extinction risk under different management scenarios, with projections over a short time period (100-years). *Linkie et al. (2006)* also used this approach for a conceptually similar analysis of tiger population viability in the Kerinci Seblat region of Sumatra. Thus the conclusion of *Armbruster, Fernando & Lande (1999)*, that examining population persistence over a 100-year time frame seriously underestimates the absolute risk of population extinction for species with long generation times (such as elephants) over a 1,000-year period, is not pertinent to this analysis.

The results of even our most optimistic scenarios are alarming, since relative extinction risks are very high even when rates of elephant removal are very low or low, with local extinction likely to occur in less than three elephant generations. Moreover, the results of other scenarios judged to be realistic suggest that local extinction is likely to occur

within 1–2 elephant generations. Thus, the ERL population appears not to be able to sustain any level of removal for translocation or indeed anything other than occasional poaching. Furthermore, if we consider the quasi-extinction scenarios (reduction to <30 or <50 individuals), which of course result in much more rapid crossing of quasi-extinction thresholds, it is clear that the ERL elephant population is likely to lose much of its social integrity and cease playing a significant ecological role in a relatively short time (potentially <15 years; baseline scenario with high removal and quasi-extinction at 50 individuals) unless a no-translocation management policy is implemented.

## Management implications
### Moving away from translocation of elephants for managing human–elephant conflict (HEC) in the National Elephant Conservation Action Plan (NECAP)

Our results suggest that Malaysia has to move away from translocation as a major method for managing HEC in Peninsular Malaysia, except in the case of 'doomed' individuals or herds (e.g., very small numbers of elephants that are isolated from other elephant populations and which may also have a highly-skewed sex- or age-structure and/or are in areas of habitat scheduled for complete conversion to other land uses). Translocation of such doomed individuals or herds to protected areas will in some cases be the only appropriate management strategy, and is the strategy recommended in the National Elephant Conservation Action Plan (NECAP), which DWNP prepared with the Wildlife Conservation Society–Malaysia Program and other partners, and which was launched officially in November 2013. More generally, the NECAP calls for elephant conservation in Peninsular Malaysia to be governed by the following principles: (1) promotion of human–elephant coexistence; (2) restoration and maintenance of socially and ecologically functional elephant population densities; (3) an emphasis on maintaining the species' present geographical range; (4) management of the CFS as three Managed Elephant Ranges (MERs); and (5) an emphasis on monitoring and adaptive management to help ensure the plan is implemented successfully. The MER concept provides a landscape-level approach in which planners assess the habitat requirements of elephants over large areas and allow for compatible human activities such as reduced impact forestry, slow rotation shifting cultivation, and controlled livestock grazing in some zones. MERs are typically established outside of—usually as extensions to—existing protected areas, and as such often include habitat corridors linking protected areas. The MER concept is particularly attractive, and probably has the greatest potential, where protected areas consist primarily of steep hilly terrain or are small and the surrounding areas are disproportionately important to elephant populations but contain agriculture or villages (*McNeely & Sinha, 1981*; *Olivier, 1978*; *Santiapillai & Jackson, 1990*).

### Non-translocation-based approaches to managing HEC and the need for research on elephant movements

For the ERL, the new NECAP approach includes explicit recognition that the area's elephant population cannot sustain even very low levels of translocation, as we demonstrate in this paper, and so other means of preventing HEC or mitigating its effects will be needed. For large commercial plantations, a non-translocation approach to managing HEC is

likely to require the use of physical barriers such as fences. Thus, it will be necessary to construct (or improve existing) barriers, especially high-voltage, well-designed, and above-all well-maintained electric fences. Use of electric fences around privately-owned cultivated lands has achieved notable successes compared to government-owned electric fences in India (*Nath & Sukumar, 1998*), while a success rate of 80% has been reported for electric fences around oil palm and rubber plantations in Malaysia (*Sukumar, 2003*). Nevertheless, the use of fencing for wildlife management has attracted considerable controversy in recent years (*Creel et al., 2013*; *Packer et al., 2013*; *Pfeifer et al., 2014*; *Woodroffe, Hedges & Durant, 2014a*; *Woodroffe, Hedges & Durant, 2014b*), in part because of the inherent risks of population fragmentation. Thus, if more widespread use of effective barriers to elephant movement is not itself to pose a threat to the elephant population by, for example, trapping elephant groups in areas too small to support them, it will be necessary to position the barriers taking elephant habitat requirements and ranging behavior into account. This will entail using data on elephant movements collected using satellite telemetry (i.e., GPS collars) and fortunately a large dataset on elephant movements in Peninsular Malaysia is now available (*de la Torre et al., 2019*).

The telemetry-based data on elephant ecology and behavior will also greatly assist with the Malaysian Government's plans to maintain elephant habitat connectivity throughout the CFS, and ultimately to re-establish gene flow between the major elephant populations within the CFS, since the study will allow critical areas for elephants to be identified and thus facilitate 'elephant-friendly' land use planning.

In addition, the needs of villagers must not be forgotten, as their small plantations and other agricultural areas are also affected by HEC. Prevention and mitigation of HEC at this scale will require a combination of community-based crop guarding methods such as simple alarm systems and village crop defense teams (*Fernando et al., 2008*; *Osborn & Parker, 2002*), the application of which has resulted in notable successes in parts of Asia (*Davies et al., 2011*; *Gunaryadi, Sugiyo & Hedges, 2017*; *Hedges & Gunaryadi, 2009*) and possibly also electric fencing around particularly vulnerable areas (rather than fencing the entire elephant habitat–agriculture interface). Again, it will be necessary to position any barriers to elephant movements taking elephant habitat requirements and ranging behavior into account, something that is often insufficiently recognized as being necessary.

### The need for law enforcement efforts to be increased

Finally, while our PVA results show that the ERL elephant population cannot sustain even low levels of removal for translocation they also show that it is equally vulnerable to even low levels of poaching. This can be seen by simply treating the translocation-related removals we modeled as deaths due to poaching because, as already noted, the underlying model structure and thus the results are the same. Moreover, even in the scenarios (including those in the sensitivity analyses) which included no translocation-related removals, population growth rates were still very low or, in some cases, negative, suggesting that management aimed at reducing elephant mortality rates is needed. Clearly, then, law enforcement efforts including anti-poaching patrols will be needed in order to protect both the ERL elephants from illegal killing (including retaliatory killing resulting from HEC, accidental deaths

due to snaring, and poaching for ivory) and their habitat from encroachment and other threats. All law enforcement work and reporting thereof should be to internationally-agreed standards (*Appleton, Texon & Uriarte, 2003*; *Stokes, 2012*).

## CONCLUSIONS

The Endau Rompin Landscape (ERL) elephant population is of clear national and regional significance, and with effective management elephant numbers could double. It is however currently of a size that makes it highly vulnerable to even low levels of illegal killing or removal for translocation. Management of the population in the future should therefore focus on (1) non-translocation-based methods for preventing or mitigating HEC including well-maintained electric fences and other deterrents to elephant incursions positioned using data on the elephants' ecology and ranging behavior; (2) effective law enforcement to protect the elephants and their habitat; and (3) efforts to maintain elephant habitat connectivity between the ERL and other elephant habitat within the Central Forest Spine.

## ACKNOWLEDGEMENTS

We thank Mike Meredith for assistance with the dung survey analyses (by helping to provide training to country program field staff in statistical methods) and Peter Clyne, Ahimsa Campos-Arceiz, and three anonymous reviewers for helpful comments on an earlier version of this paper. We are grateful to the DWNP's Elephant Unit for sharing their knowledge of translocations and HEC and WCS-Malaysia Program's Elephant and Tiger Projects' field staff for helping with surveys. Finally, we acknowledge the support of the Dato Sri Douglas Uggah for promoting in situ conservation of elephants in Malaysia.

### Funding
Funding for this project was provided by the Department of Wildlife and National Parks, the Johor National Parks Corporation, the Wildlife Conservation Society (WCS), the U.S. Fish & Wildlife Service's Asian Elephant Conservation Fund (No. 98210-7-G198), the CITES Monitoring the Illegal Killing of Elephants (MIKE) program (No. QTL-2234-2661-2310-211200), and the Denver Zoological Foundation, with in-kind contributions from the State Forestry Departments of Pahang and Johor. The funders had no role in study design, data collection and analysis, decision to publish, or preparation of the manuscript.

### Grant Disclosures
The following grant information was disclosed by the authors:
Department of Wildlife and National Parks.
Johor National Parks Corporation.
Wildlife Conservation Society (WCS).
U.S. Fish & Wildlife Service's Asian Elephant Conservation Fund, the CITES: 98210-7-G198.

CITES Monitoring the Illegal Killing of Elephants (MIKE) program: QTL-2234-2661-2310-211200.
Denver Zoological Foundation.
State Forestry Departments of Pahang and Johor.

## Competing Interests

Melvin Gumal is employed by the Wildlife Conservation Society (WCS); Simon Hedges, Aris Oziar, and Martin Tyson were employed by WCS at the time of the study. Salman Saaban is employed by the Department of Wildlife and National Parks (DWNP) in the Ministry of Water, Land and Natural Resources; Abd Samsudin and Mohd Nawayai Yasak were employed by DWNP at the time of the study. Francis Cheong was employed by Johor National Parks Corporation (JNPC) at the time of the study. Zaleha Shaari was employed by the Department of Town and Country Planning (DTCP) at the time of the study.

## Author Contributions

- Salman Saaban conceived and designed the experiments, performed the experiments, authored or reviewed drafts of the paper, government oversight of study; discussion of policy and wildlife management implications, and approved the final draft.
- Mohd Nawayai Yasak authored or reviewed drafts of the paper, government oversight of study; discussion of policy implications, and approved the final draft.
- Melvin Gumal conceived and designed the experiments, performed the experiments, analyzed the data, prepared figures and/or tables, authored or reviewed drafts of the paper, coordinated NGO involvement in study, and approved the final draft.
- Aris Oziar conceived and designed the experiments, performed the experiments, analyzed the data, prepared figures and/or tables, and approved the final draft.
- Francis Cheong performed the experiments, prepared figures and/or tables, authored or reviewed drafts of the paper, discussion of policy implications, and approved the final draft.
- Zaleha Shaari prepared figures and/or tables, authored or reviewed drafts of the paper, discussion of policy implications, and approved the final draft.
- Martin Tyson conceived and designed the experiments, performed the experiments, analyzed the data, prepared figures and/or tables, authored or reviewed drafts of the paper, discussion of policy and wildlife management implications, and approved the final draft.
- Simon Hedges conceived and designed the experiments, performed the experiments, analyzed the data, prepared figures and/or tables, authored or reviewed drafts of the paper, and approved the final draft.

## Field Study Permissions

The following information was supplied relating to field study approvals (i.e., approving body and any reference numbers):

The work was approved by the Department of Wildlife and National Parks (DWNP), Kuala Lumpur, Malaysia and Johor National Parks Corporation (JNPC), Johor, Malaysia; senior officials from both organizations participated in the work and are authors.

## Data Availability

The raw data are available in the Supplementary Files.

## Supplemental Information

Supplemental information for this article can be found online at http://dx.doi.org/10.7717/peerj.8209#supplemental-information.

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
