# Peer review of "Viability and management of the Asian elephant (Elephas maximus) population in the Endau Rompin landscape, Peninsular Malaysia"

_PeerJ, doi:10.7717/peerj.8209_

## Round 0.1 · original submission · Minor Revisions

Thank you for this well-written and well-executed study. Please consider making some minor edits to improve clarity, based on the three thorough reviews received below. When submitting your revised manuscript, please include a response to reviewers summarizing your edits.

Thank you for submitting to PeerJ.

Reviewer 1 ·

Basic reporting

This article is clear and logical with good references.

Some additional information is needed to support the claim that the authors worked in close collaboration with policy makers (see my comments to the authors). Some literature cold be updated (see comments)

Experimental design

No particular concern on this.

Validity of the findings

the major finding that translocation could have harmful effects on the viability of the elephant population is very important to stress out. Data are quite robust although poaching seems to have been underestimated as a cause of extinction by the authors. It would be important to give more details about what should be the priority actions to be undertaken to save this population from extinction.

Additional comments

The authors bring some very interesting information about the elephant poulation living in Endau Rompin landscape. The point made by the authors to stop translocation is very important to make, and I applaud the authors of this study to bring this element to the knowledge of land deciders and wildlife managers in West Malaysia. This conclusion may have very serious consequences for managing elephant populations, not only in Malaysia but across Asia. This fact is crucial to be published and to be brought to the general knowledge.

I have made few minor comment below.

Introduction L59-64: the statistics given by the authors are from 2005 and 2011. More recent statistics are available and should be used instead.
L64-67: needs for a reference
L 82-89: I’m surprised that only two elephants have been monitored with satellite tracking over the years. Is there any reason why not more elephants were monitored using this technology?
L108-110: the fact that there is no size estimate for this population should be highligted by the authors as one of the reasons for their study.
L 116-119: what is the "question of immediate relevance to conservation in Malaysia” that the authors want to adress in their paper? It needs to be clearly stated.
L 275-290: I don’t discuss the importance for conservationists to work more closely with field practitioners and land deciders in order to inform startegies for land uses, land management and else. So I fully agree with what the authors write in this section. However, the authors should be more precise and they should detail what they did to justify their statement “this study provides an example of scientists working with policy makers…. (L 285-290)". Indeed, the authors don’t show anywhere that they engaged with policy makers or field practitioners during their study. Have they organized workshops, or discussions or consultation of any kind? Have they identified the research question following meetings with relevant authorities? And so on. The authors mush bring the evidence that indeed, they collaborated with different stakeholders during their study.
L302-304: I agree that the lower density may result from emigration. Although I imagine that the number of transocated elephants from this population is well known (and in this case, it would be very interesting to give the figure and the time frame), it is most probably more difficult to know the details about killing. However, killing could be even more of a problem knowing that many elephants in this population are shot for their ivory or during HECs. Although the Vortex includes some level of killing, the authors should emphasise (if this is the case) that killing is a real threat to the long-term viability of this population and that everything needs to be done to stop this. Both transocation and killing are part of the same problem: excessive emigration from the source population, as the authors write in section 410-421. It could be explained earlier than in this later section that killing and translocation have the same impact on a population viability.
L 408-421: it would be interesting for the authors to give more information about the level of hunting witnessed by this population.

Reviewer 2 ·

Basic reporting

General comments

Overall, the manuscript is very well written, with clear and unambiguous English used throughout.

Seems that the manuscript was written years ago:
Lines 62-63: ‘with about 28% of the peninsula projected to be under these crops by the end of 2015’. I recommend rewording this sentence and if possible using actual data on the crops area coverage.


The article is generally well structured. Below are some comments by section:

Introduction
Introduce the Central Forest Spine (CFS) master plan in the introduction. The CFS is presented in the discussion but I think it is highly relevant and it would be better to introduce it earlier on.
In Methods (line 194) it is mentioned that the population in Endau-Rompin Landscape (ERL) is a closed population. That is very important information, I recommend mentioning it in the introduction. In the discussion, I recommend discussing the implications of lack of connectivity in the context of CFS and the Managed Elephant Range. Although there is a brief mention in lines 311-314, it would be good to provide more details on how challenging would be to reestablish connectivity with other populations.


Methods
Study Area
I recommend to add a bit more of detail on the study area, specifically on the land use and vegetation types. This is important to understand the potential elephant densities. Sukumar (2003)’s suggestion of 0.1/sq-km refers to old-growth continuous forest. If the forest is highly fragmented, has been selectively logged, or includes crops like paddy fields, this would have implications for potential elephant densities.

Also, in line 223 it is mentioned that some dung piles were destroyed by construction work within the study area. Can you please mention in the Study Area section what kind landscape changes were taking place at the time of the study?

Population Viability Analysis
Can you provide any real data on the number of elephants translocated from ERL. Also, can you please indicate where they are translocated to? As far as I know, in the past few years elephants from the State of Johor were released within ERL but in line 303 it is mentioned that some elephants were translocated out of ERL. Some more details would be helpful.

Discussion
Need for research on elephant movements => some work has been done on elephant movements in this landscape. The authors can contact me ([email protected]) to discuss potential collaborations regarding elephant movement data. Actually, the study proposed in lines 391-395 already exists and is currently under review (I’ll be happy to share the manuscript with the authors if they are interested).

Experimental design

The methods are appropriate for the purpose of the study

Validity of the findings

The findings are valid and relevant for conservation

Reviewer 3 ·

Basic reporting

The paper is very well written, structured and supported by the appropriate literature. A very enjoyable and easy read. Obviously this paper has undergone a few reviews. However, I did find one little annoying language issue; the excessive and incorrect use of semi-colons and colons. I have highlighted in the attached: examples of where they are used incorrectly. In most cases a comma could be used or perhaps a new sentence. Finally, the abstract needs a little work. The third sentence is a little long and fragmented. The other issue is the excessive number of tables and figures.

One area that needs improvement are the figures and tables.

Figures 3 to 14 could be condensed into figure panels or included in supplementary. Also, it would be good to include a letter i.e. (a), (b) etc. and describe what each plot means. The captions are also a little bit hard to read. They have strange formatting with a carriage return in the caption. My suggestion would be to put all the sensitivity analysis figures in the supplementary. This paper really about the PVA itself and the modelling methods and they barely get a mention in the text; so I don't think they need to be in the actual manuscript.

The tables also are excessive - too many. And obviously they are not an important part of the story as they are rarely mentioned in the text. Again I think they could be moved to the supplementary.

I think the authors have to figure out some way to summarise some the data in the figures and tables into a plot or table. Perhaps only plot the PE and have different curves for different paramaterisations.

Experimental design

no comment.

Validity of the findings

The findings appear valid to me.

Additional comments

In general this was a well executed study. The research fills a very important research gap. The application of PVAs in Southeast Asia (and the global south in general) is rare, even though there are such pressing biodiversity issues which can be addressed with PVAs. In general I am happy with how the analysis has been presented and the application is rigorous and of high technical standard.

External reviews were received for this submission. These reviews were used by the Editor when they made their decision, and can be downloaded below.

---

## Round 0.2 · accepted · Accept

Thank you for your revisions.

External reviews were received for this submission. These reviews were used by the Editor when they made their decision, and can be downloaded below.